# (Inter)racialization: the regulation of domestic and urban space in housing North African migrants in 1960s and 1970s France

Franco, Rébecca S.[1],

**1** VU University Amsterdam, Faculty of Law
* r.s.franco@vu.nl

Monday, February 14, 22

## Abstract

**Literature on immigrant housing and assimilation has shown how housing policies perpetuate, create, and contest racialized boundaries. This paper argues for the necessity to look at regulation of the domestic space together with regulation urban space. By reading "along" and "against" the archival grain of the French national archives and the Paris city archives, this paper looks at housing policies that targeted the North African migrant population in the 1960s and 1970s in France as colonial continuities. French authorities ostensibly encouraged gendered assimilation through spatial politics and interventions in the domestic space. Literature on the French context has argued how this perpetuated racialization in the housing process. Building upon feminist scholarship on gender, intimacy, and colonialism, this paper shows how these policies negated interracialized households and prevented interracialized intimacies. This helps understand how housing policies can reinforce racialized exclusion by regulating racial boundaries in urban space and domestic space together to not allow for and prevent interracialized households.**

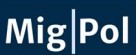

## 1.     Introduction

Today, the French public authorities see the *banlieues spaces* in the urban peripheries in France through a prism of problems of segregation and integration of their racialized inhabitants, mostly (descendants of) people coming from the (former) colonies on the African continent (Dikeç, 2011; B. S. Epstein, 2011). Contemporary urban politics enforces spatial distribution through social mixture policies (Lefevre, Roseau, & Vitale, 2013), but ambiguous and uncertain goals lead to conflicting implementation that contributes to persistence in inequalities (Boisseuil, 2019). Scholars in Western Europe more generally have explored whether and how discourse and practice on spatial politics and migrant integration work to exclude those who are to be integrated (Bolt, Özüekren, & Phillips, 2010; R. Epstein & Kirszbaum, 2003; Musterd, 2003; Schinkel, 2013, 2017, 2018). The French case in particular is interesting because of the French universalist investment in colorblind politic, which scholars have argued perpetuate racial formations and inequalities rooted in colonial histories (Beaman & Petts, 2020; Fassin & Fassin, 2013; Stovall & Van den Abbeele, 2003). Looking at the histories of the housing projects that created the banlieues in de 1960s and 1970s can help understand the construction of racial projects in France, by which I refer to Omi and Winant's understanding of racial projects as governmental practices and policies that bring together ideological and material aspects of race to organize and distribute resources and capital along racial lines (Omi & Winant, 2014).

This paper looks at the housing policies that targeted migrants from the former colonies during and in the wake of political decolonisation. The French administration problematized and regulated migrants within French metropolitan territory through housing policies, based on colonial modes of governance (Bernardot, 2008; Hajjat, 2018; House & Thompson, 2016; Lyons, 2006; MacMaster, 1997). Researchers have shown that French authorities implemented housing policies that were concerned with gendered assimilation and spatial distribution, but enforced exclusion of migrants from the French community (F. Belmessous, 2013; Bernardot, 2008; Hajjat, 2018; Hmed, 2006; Lyons, 2006, 2009, 2013). However, this scholarship does not look at how the household categories are made in the first place. To more fully comprehend the intent and outcomes

of these policies, this paper looks at how categories are crafted through the interventions in domestic space, and how this orders urban space.

This paper traces colonial continuities in the regulation of urban space and domestic space together to better understand the ways in which housing politics produce and undo racial boundaries. To do so, I build on insights from feminist research on gender, intimacy, and colonialism. This scholarship argues that the regulation of "sexual, conjugal and domestic life" and interracialized intimacies were essential to the colonial order of things and the protection of racial hierarchies (Camiscioli, 2009; McClintock, 2013; Povinelli, 2006; Stoler, 1989, 2010). I thereby understand the domestic space as referring to the space that belongs to the household, crafted through state interventions. The concept alludes to the French term "*domestiquer*": to domesticate, to subjugate a population to colonial power. To do so, I look at the interracialization of domestic space and at interracialized intimacies. I use the concept "interracialization" to refer to the process of assigning different racialized identities to members of the household. Interracialized intimacies refer to the intimate relationships (sex, marriage, unmarried relationship/cohabitation) between people who are assigned different racialized identities. I base this on 'interraciality' used in critical (mixed-)race studies to refer to the construction of interracial relationships and families/couples through the law, politics, and discourse (Ifekwunigwe, 2004; Onwuachi-Willig, 2013; Onwuachi-Willig & Willig-Onwuachi, 2009). Building on Haritaworn's work on "multiracialization", I underline the process involved in racialization, and therefore, in interracialization (Haritaworn, 2007).[1]

I apply this understanding of interracialization and interracialized intimacies to look at how housing policies crafted the domestic space and urban space in a way that did not acknowledge interracialized households and prevented interracialized intimacies all the while promoting assimilation. In doing so, this paper adds to the literature specific to French housing policies to show how these regulated interracialization. It thereby also adds to literature on migrant integration and spatial politics more generally to argue for the necessity to look at the production of urban space and domestic space together.

I will first go into the methods on which this paper is based. Then, I will set out the context in which the housing policies developed, to show how the tools of governance were colonial continuities that travelled from the colony to the metropole. Building upon these

colonial continuities, I show how the housing policies consolidated and were based on racial and gender hierarchies of assimilability that connected the regulation of urban space with the regulation of domestic space. Looking at these connections reveal how housing policies regulated interracialized intimacies.

## 2.    Methods: tracing racialization in the archives

This paper is based on primary and original archival sources from the National Archives of France and the archives of the City of Paris. I consulted these archives in the context of my doctoral research on the regulation of interracialized intimacies between the white French population and North African immigrants between 1956-1979. I selected archives based on key-word searches concerning African migration and went through the physical and PDF inventories compiled by archivists on certain state services, ministries, or themes to obtain a comprehensive overview of the structure of the inventory and the archive. After having set out the basic understanding of the different policy-fields that affected interracialized intimacies, I proceeded with more targeted searches on housing and social action for this particular paper. The archives include those of the Ministry of Construction, the Ministry of Housing, the Ministry of Interior Affairs, the Ministry of Labour, and the Ministry of Health, and the Municipality of Paris. I used freely available material and material for which I have received special permission to consult [*derogation*] under the *213-2 du Code du Patrimoine*. I have looked at governmental policies, correspondence, research, circulars, and legislation.

Research on race and racialization of regulation in France comes with its own set of challenges, as the French tradition of colorblind universalism does not acknowledge the existence of racialized processes and governance. However, at the same time, research on race and racism in France has argued that racial logics are embedded within the French Republic – yet never made explicit (Célestine, 2011; Stoler, 2011; Stovall & Van den Abbeele, 2003; Thompson, 2016).  Stoler calls on researchers of racial histories in France and elsewhere to "*ask who and what are made into 'problems', how certain narratives are made 'easy to think', and what 'common sense' such formulations have fostered and continue to serve*" (Stoler, 2011).

Gordon has argued that regulations do not (only) impose things, but rather dispose things in a certain way by presenting it as the "natural order of things" (Gordon, 1984).Categorizations are integral to problematization because they form the discursive and regulatory basis on which certain groups become defined as a problem, and create the identities that can be regulated (Grommé & Scheel, 2020; Schrover & Schinkel, 2013). As different governmental rationales on assimilation and immigration lead to different policies and policy-outcomes (King, Le Galès, & Vitale, 2017), it is important to look at both the governmental policies and categories and the omission of certain policies and categories to understand how the state "disposes an order of things". Building on this understanding, I look at how regulation, categorization, beaucratic silences and ommisions disposes a racialized order as the natural order of things.

For this paper, I approached the archive with the question of whether and how French authorities were interested in the regulation of the intimate lives of the (North) African population in the French metropole, and specifically, whether and how the authorities were interested in interracialized intimacies. I looked at the archive as an "object of knowledge" rather than a "source of knowledge" (Arondekar, 2005; Stoler, 2002). Scholars have argued for a reading "against the archival grain", in which the researcher attempts to uncover that what is not being said, those knowledges that are disqualified (Burton, 2006; Whatley & Brown, 2009). Stoler has famously argued for a "reading along the archival grain": a reading that treats the archive as a "force field" to which the research should surrender (albeit not concede) to trace its logics, to show what and how governmental rationales order governance. In this paper, I do both: by placing different types of archives and different sources next to one another I could trace and (re)interpret the silences, hypervisibilities, invisibilities, fragmentation, inconsistencies, and assumptions on categorization and regulation of (North) African migrants. In doing so, I am able to show the underlying ordering principles of the regulation of domestic and urban space.

## 3.     Colonial continuities

This paper looks at the housing policies that targeted postcolonial immigrants from the former colonies in North Africa during and after (political) decolonization of the French

Empire, and specifically Algeria, between 1960-1975. This period is marked by rapid change and economic prosperity, yet with housing shortages (not unlike other European states at the time) (van Beckhoven, Bolt, & van Kempen, 2009). From the second half of the 1950s onwards, France waged a war against the nationalist movement of French Algeria. The war culminated in the eventual independence of Algeria in 1962, following a period of intense state violence against Algerians in both French Algeria and the metropole (Brun & Shepard, 2016).

Around this same time, the French state was interested in the modernization of its country through urban renewal and mass consumption in the wake of the Second World War, for which they necessitated cheap migrant labour (Ross, 1996). In this context, the French state implemented free movement between colonies and former colonies on the African continent, up until the changes in immigration legislation in the mid- 1970s that were the consequence of the economic downturn following the Oil Crisis in 1972 (Weil, 1995). In reality, however, the French administration developed administrative regulations to limit and control migrants from North Africa on French territory (Laurens, 2009; Spire, 2005).

Understanding colonial continuities in the regulation of North African migrants helps better understand the specialized housing policies for migrants. Scholars has shown that the French administration regulated migrants from Algeria separately through specialized institutions and categories that created intimate connections between colony and metropole, and between the colonial and postcolonial context (de Barros, 2003, 2005, 2006; House, 2004; House & Thompson, 2016; Lyons, 2006, 2009, 2013; Naylor, 2013). Before independence of Algeria, Muslim Algerians were French citizens – albeit regulated under the legal regime of "Muslim status" which de facto created a differentiated set of rights in French Algeria (Blévis, 2004). Officially, French Algeria was a French department of the metropole and therefore administered by the Ministry of Interior Affairs.

In the metropole, Muslim Algerians had, at least formally, no different status to French citizens. Still, French metropolitan officials used the category of *Français Musulmans d'Algérie* [French Muslims of Algeria – henceforth Muslim Algerians] (FMA), which allowed the administration to differentiate between them and the rest of the French citizenry (Lyons, 2013, p. 24). The usage of distinctive categorizations for Muslim

Algerians made possible the creation of specialized institutions and services that obfuscated the lines between colonial and metropolitan government. The department of Algerian Affairs of the Ministry of Interior opened up branches and instated specialized services in the metropole to regulate Muslim Algerians (de Barros, 2005). Through these institutions, colonial practices of governance, individuals and expertise travelled between French Algeria and metropolitan France. These interconnected housing with social welfare, surveillance and assimilation to monitor and regulate the colonial (migrant) populations.

Housing shortages were an acute problem in France, but the state investment in migrant housing went beyond a sole interest in the improvement of living standards: housing was one of the sites through which the French administration codified classifications, problematized some groups of migrants, and managed migrants in the French territory (Blanc-Chaléard, 2016; de Barros, 2005). The poor housing in which many workers from the African continent were living, were seen as an aberration to the modern French state (Blanc-Chaléard, 2016). The administration developed housing policies that targeted the FMA category specifically, and separately from the French population, as I will show throughout this article. Such efforts took force in the Algerian War, to suppress nationalist sentiment amongst the Muslim Algerian population and promote a "civilizing mission in the metropole" (Lyons, 2013). Throughout the 60s and 70s, French authorities continued to worry about existence of *logements insalubres* [unsanitary housing] and *bidonvilles* [shantytowns]. This fuelled anxieties about the presence of migrants from the African continent on French territory altogether.

Metropolitan practices targeting North African migrants should be seen as colonial continuities of technocratic governmental techniques for modernization and rationality (Laurens, 2009; McDougall, 2018; Rabinow, 1995). In French cities in North Africa, such as Algiers and Casablanca, the colonial administration implemented housing policies that deconstructed social structures of the "Muslim city" to better surveil and manage the Algerian population as part of the colonial project of the *mission civilatrice* [civilizing mission] (Çelik, 1992, 1997; Rabinow, 1995; Wright, 1991). In the post-independence context, colonial institutions and officers who had been mandated with the regulation of the Muslim population in French Algeria and the French metropole were rebranded to manage the foreign immigrant population. They brought with them spatial politics and

urban planning and the crafting of domesticity as a colonial tool of governance (Almi, 2002; de Barros, 2005; House, 2018; Wright, 1987).

In doing so, the policies intervened in both domestic space and urban space. Colonial officers and architects built differentiated housing units for the Muslim population that were supposed to contribute to the so-called evolution of the Muslim population into modern French city life, by building European-style housing catering to and crafting a nuclear family. The colonial administration attributed such housing units based on the level of the so-called evolution of the family, by measuring the behaviour of the woman. They thus approache the "Muslim woman" as the key by which the "Muslim home" could be controllably assimilated into French domestic life (McKay, 1994). Being attuned to colonial continuities thus points to the necessity to look at how the crafting of domesticity has been at the centre-point of imperial politics and the separation between colonizer and colonized (Conklin, 1998). This focus on gendered assimilation as having both a spatial and domestic element sheds light on the housing policies targeting postcolonial migrants in the 1960s and 1970s.

## 4.      Hierarchies of assimilability

Before looking at housing policies, it is helpful to look at how the administration constructed hierarchies of assimilability that formed the basis of the housing policies. In the period after independence and before the crackdown on immigration in the mid-1970s, the French administration allowed free movement from Algeria. However, they favoured European migrants over North African migrants, motivated by concerns on assimilability packed in technocratic arguments (Laurens, 2009; Spire, 2005; Weil, 1995). The French administration asserted that European migrants and African migrants were inherently different because the former could assimilate into the white French community, but the latter could not. This is stated throughout reports on migration. For example, the Institute of applied research on housing report commissioned by the Ministry of Housing in 1968 stated that "the Spanish and the Italians assimilate easily: their arrival is desired", but that it is mostly "Maghrebi, especially Algerians" whose adaptation caused problems. Whereas prior to Algerian independence, concerns revolved mostly around Algerians, after independence the authorities and politicians used the

category of "North African" or sometimes "Maghrebins" to refer to all migrants from North Africa, such as Tunisia and Morocco. In the postcolonial metropolitan context, taxonomies of assimilability thus justified racialized differentiation, loosely and inconsistently referred to by nationalities or geographical regions.[2]

The notion of the inassimilability of North Africans is rooted in French colonial racial hierarchies. Research has shown that racism in France continues to target Muslim citizens and migrants (Hajjat, 2012; Mayer, 2012, 2018). The colonial government used the ideal of assimilation to promise colonial subjects the possibility to be granted full rights as French citizens – and therefore worked to uphold Republican universalism (S. Belmessous, 2005). This, however, was racially circumscribed: assimilation was a goal that could (almost) never be fully attained for colonial subjects, and therefore worked to exclude racialized colonial subjects and uphold the colonial order (Coquery-Vidrovitch, 2001). Moreover, research on the metropolitan context of the interwar years shows that the French administration encouraged a racialized understanding of assimilation in the regulation of immigration, out of eugenic demographic concerns, that favoured European migrants over colonial migrants (Barton, 2020; Camiscioli, 2009). These racialized understandings of inassimilability thus travelled from colony to the metropole with the arrival of colonial migrants.

The problematization of gender and marital status played an important role in the construction of North African migration as inassimilable. North African migration was understood as single, temporary, and male low-skilled and low-class labour migration, as opposed to European migration. Authorities regulated immigrant workers from the African continent based on their capacity as male workers. Policy documents use the category "*isolés*" (isolated) to designate single men who came from the African continent to work in France, which informed policy-making. For example, the Minister of Interior stated to the Prime Minister in 1963 that migration from the African continent is only temporary and argued that it was therefore in no one's interest to facilitate permanent residence. The letter reiterated multiple times that this immigration is temporary.[3] As the Algerian sociologist Sayad has argued, African male migrants were seen as having an "in-between" family status. They were neither really married, even if they have families in their home country, nor really without family. This placed them outside of the French

conception of the nuclear family (Sayad, 1980, 1997) – thereby marking them as outside of the French community.

Even though authorities and policymakers understood family migration as facilitating assimilation of so-called 'men without families', the French administration worried about North African families' arrival because it would bring about the risk of durable installation of Algerian families (Cohen, 2017). In contrast, French authorities encouraged the family migration of European migrants, such as Italians. because they were considered to be assimilable into the white French community (Cohen, 2014; Spire, 2005). This informed the French administration's position on immigration from the African continent, as illustrated by a report from 1966 report from the social services mandated with social action for migrants.

> Migrants from European countries: Italians, Spaniards and Portuguese with a behaviour that gives them the best chance of integration into the national community […] Migrants from North Africa and Black Africa have a relatively high number of a-socials whose adaptation seems, a priori, excluded. […] This population is mainly composed of single working men. The presence of families, an element of stabilisation, is particularly lacking.[4]

The focus on single men's temporariness enabled the authorities to negate the possibility of settlement, justified the discouragement of family migration and motivated inassimilability of North African migration.

However, the construction of North African migration as temporary and unassimilable did not acknowledge the reality of the presence of interracialized couples and households. Governmental statistics in the archives make it difficult to recollect the presence of interracialized households. In explicitly looking for traces of their presence in the archives, however, I was able to find that a part of the households categorized as "North African" or "Muslim" consisted of a woman that was white French or European. The trimestral reports by the CTAM give statistics on the amount of "European wives and concubines" in the "Muslim population" between 1959 and 1964: in '63, the CTAM counted 10 700 "European wives and concubines" and 36 000 "Muslim wives".[5] The organisation "Entraide Nord-Africaine d'Indre et Loire" counts in 1961 700 North African families and 240 mixed families in the department of d'Indre-et-Loire – about one fourth.[6] The Direction of population and migration of the Ministry of Social Affairs

counts about 57 000 Algerian families in France in 1968, of which 52 000 have an Algerian national as "head of household". So, about 5 000 Algerian families had an Algerian mother with a non-Algerian head of household.[7] Of the "Algerian head of households", 17 000 are married to French women. In 1975, the General census of the Population counts 92 000 Algerian men in a relationship, of which 24 000 live with a French woman. The statistics of the census of '68 and '75 count based on nationality, meaning that men born Muslim Algerian but who have been naturalised as French are not counted in the statistics – thereby invisibilizing interracialized households where both partners have French nationality. However, social housing officials still considered North African families who had been naturalised as French as North African. These statistics show that almost one in four families counted as "North African" were interracialized.

Yet, most of the reports and policy documents on housing and social action do not mention the presence of French or European women. Instead, they crafted North African migration as unassimilable and problematic, justified by the notion of assimilation that especially targeted women. The social action reports even reported on the type of clothing of Algerian women: dressing "*a la francaise*" was a positive marker of adaptation.[8] Educational courses taught women domestic skills: how to sow, how to clean their houses, how to rear their children, how to be proper French wives.

However, social action negated the presence of French wives. For example, a report of the branch of the SLPM mandated with housing and social action for North African families in 1965 did not mention prevalence of families that consisted of a North African man and a wife that was born and educated in France was not mentioned as a degree of assimilation, even though between one third and one-fifth of the families/couples classified as North African at the time consisted of a French or European (white) partner.[9] In contrast, the authorities praised the prevalence of marriage between French women and European migrants as a sign of assimilation.[10] This is similar to the interwar years (Barton, 2020). As I will show, the invisibility of French wives and the prevention of interracialized intimacies more generally was justified ánd was a consequence of the institutional differentiation of North African migrants in the housing policies, and the separation of single men from families.

## 5.      Racialized difference: segregation and cohabitation in urban space

The French authorities developed housing policies that consolidated hierarchies of assimilability, determined by logics on race, gender, class, and marital status that did not allow for the possibility of interracialized households. Legislation and policies differentiated between "housing for the isolated" – isolated being the term employed by the administration – and "housing for families".[11] The authorities built segregated communal housing for single men motivated by gendered exclusion – as I will show below. At the same time, the French authorities justified specialized housing for North African families through a logic of gendered assimilation in urban and domestic space.

The administration built two types of housing that housed North African families: *cités de transit* [transit centres] and *habitations a loyers modéréees (HLM)* [social housing]. Officially, migrants were eligible for regular HLM housing and had to be considered without difference from the French population. However, HLM bureaucrats used aminimum living requirement for ten years to exclude migrants from HLM housing (David, 2016). Consequently, most North African families could only accede to HLM housing in the context of so-called "slum clearance"[12], which often meant that they had to pass through the *cités de transit*. Especially from 1969 onwards, the administration focused on building *cités* in their efforts for urban renewal. These specialized housing structures were based on the differentiation between North African families and French or European families, and determined spatial distribution policies.

Based on supposed inassimilability to modern French life, the cités de transit enforced urban segregation. Formally, the transit centres were built for any family who was *inadaptée* [inadapted], both French and migrant. Regulation on the transit centres also stipulated this, such as in the circular on the transit centres from 1972.[13] However, in practice, the administration did not want to house French families in *cités de transit*, even though French families were also living in unsanitary housing, and needed to be rehoused. A 1971 report of the Prefet of the Paris region on the "resorption of the *bidonville*s and the problem of migrants" asserted that housing  French families in cites de transit for migrants "should be prohibited and no exceptions should even be tolerated"[14]. The centres

functioned as a segregated space for non-white migrants rather than a centre that aimed to improve the conditions of the poor working-class. Already in 1963, research showed that the families living in the *cités de transit* were slow to integrate because of segregation. However, the failure to integrate was read as a symptom of the poor adaptability of migrant families, thereby reinforcing racialized difference.[15]

In doing so, the authorities could spatially exclude and monitor North African families and keep them in a state of temporarity. The left-wing newspaper *Liberation*, described the *cités* as "a deliberate and planned policy of deporting and locking up these sections of the population".[16] The *gérants* [managers] of the transit centres had the authority to surveil the families and intervene in their domestic space when deemed necessary. Migrant organizations and activists criticized the *cités de transit* repressive character: they argued that the *gérants* of the cités used their uncontrolled power to behave as "the king in the cité" and to rule with a "reign of terror".[17] In residents' own words "here we are secluded, we wonder if we are human or if we are taken for savage animals, savage animals that must be isolated from civilization, this is a concentration camp".[18] The temporary and repressive climate of these centres allowed the authorities to control the migrant population, and expulse those considered unwanted from French territory.[19] The transit centers also had a capitalist interest: building transitory and temporary cités also served a capitalist interest: it allowed the authorities to re-purpose the land on which the centres was built if deemed desired (Ginesy-Galano, 1984). The argument on assimilation thus served the interest of the French state.

Even though housing policies created segregated spaces by building *cités de transit*, authorities also raised concerns about the necessity to counter segregation of North African families in HLM housing and encourage so-called "*cohabitation*" between North African and French and European families. They problematized "the well-known tendency of Algerians to gather in a certain number of districts which they quickly transformed into a *medina*".[20] The use of the term medina reveals its colonial undertone. Similar to the urban planning in the colonial context in French Algeria that enforced cohabitation through special housing policies for the Muslim Algerians, administrators in the metropole argued that cohabitation was a way to insert Algerian families within the French community and calm nationalist sentiments amidst the Algerian war (Blanc-Chaléard, 2016). Discussion on the necessity of spatial distribution revolved around

concerns about, as per the terms used in the policy documents, "*cohabitation*" and "*brassage*" [mixture – lit: brewing] and segregation, and the positive and negative impact this would have on the population groups that were to cohabitate or not. This continued after Algerian independence, when the necessity to "cap" the number of North African families in a given housing estate or neighborhood became common practice.

Politicians and bureaucrats alike believed that above a certain threshold, assimilation was impossible, and the (white) French community would not tolerate migrants' presence: this understanding became popularized under the notion of the *seuil de tolerance* [tolerance threshold].[21] These concerns revolved mostly around housing, but also around schools and children's camps, and around local shops and restaurants in a given neighborhood.[22] This again had an underlying capitalist interest: officials warned that buildings "and even the entire neighborhood" depreciated in value when too many North Africans moved in.[23] Justified by the "tolerance threshold", authorities set in place in HLM housing (and not for *cités de transit)* a semi-formal system under which housing for migrant families was "exchanged" for housing for "European families" (that is: white) to avoid segregation and stay under the so-called tolerance threshold.

The authorities asserted that the quotas and the tolerance threshold should not be discriminatory, but at the same time, implemented discriminatory quotas through discretionary measures. The cap was never a mandatory policy across France, but more a rule of thumb used by officials. Whereas the discussions between high-ranked officials placed the maximum quota on fifteen per cent, in reality, only about five per cent of apartments were attributed to migrants. However, the tolerance threshold did translate into local circulars that prohibited new migrant families to live in certain neighborhoods, referred to as *medinas*, where the local governments considered that the threshold had been reached.[24] This latter circular mentioned explicitly that it is prohibited to implement "discriminatory measures that apply the rule only to Algerian, Moroccan, and Tunisian families". However, throughout the policy documents and discussions, it targeted almost exclusively these families. The authorities thus had an awareness of its discriminatory workings, but continued to implement caps that were concerned with limiting the presence of North African families in urban space under the guise of assimilation and tolerance.

## 6.    Interracialization in domestic space

Looking at policies on level of urban space together with policies targeting the domestic space shows that the segregation of families in *cités de transit* and cohabitation of families in HLM housing depended on constructing the "North African household" as a monoracialized category that could be segregated and/or spatially distributed through quotas – and controlled. Whereas research has argued that the policies that enforced cohabitation and segregation perpetuated racialization in the housing structures, this research does not take into account the presence of interracialized households (F. Belmessous, 2013; Hajjat, 2018). Interracialized households were present in both the *cités de transit* and HLM housing for North African families. However, interracialized households confound the racialization of North African households as different to French or European families, who can subsequently be spatially managed through segregation and so-called cohabitation. Looking at the household category shows that spatial distribution not only enforced racialization in housing policies, but was dependent on the regulation of interracialization.

The authorities did not respond favourably to interracialized families who made themselves visible outside of their identification as a North African family. Monique Hervo, an activist militant who lived for years in the informal settlements of Nanterre, described in her journal the experience of Jeanette, who was married to an Algerian man and arrived in the "*bidonville de Nanterre*" in 1957.[25] In 1968, after 11 years of living in a make-shift home, Jeanette went once again to the social housing services (HLM) to ask about her application for an HLM apartment. At the prefecture de la Seine, the official responsible for social housing applications responded to her demand by proclaiming that he would not help her because the HLM is "not for 'small goats'" [pejorative, racist term for Arabs][26]. He went on to exclaim that "Negros [sic], and all that, is not my area".[27] The official saw Jeanette as a North African labourer's wife and she was therefore subsumed under the North African household category. She was consequently placed outside of the Frenc community, to uphold the differentiation between North Africans and French households. The housing policies made Jeanette's situation, and thus interracialized households, unthinkable, unintelligible within the system.

The invisibilization of interracialized households should be understood as an investment in the upholding of a racialized distinction between North African and French families that justified segregation. A research carried out by the *Etudes Sociales Nord Africaines* on the North African population in the Parisian suburb Grennevilliers from 1963 shows that out of the 26 families living in the centers, three are categorized as "mixed". [28] Given that between one third and one fifth of the North African families consisted of a French white female partner, it is likely that at least some interracialized couples were also living in the *cités de transit* – even though they are not discussed in statistics or policy documents. This can be explained by the fact that North African families were excluded because they did not fit the ideal of French domesticity: the wives had to learn to lead a "proper" French domestic life before they could be assimilated into the French community: their presence did not "fit" the framing of cité de transit as a segregated space for gendered assimilation.

The existence of the *cités* was motivated by an assimilationist goal that the administration knew did not function in practice. They were built to, at least on paper, function as temporary housing for families who were considered to "not have the necessary degree of evolution" to live in a modern apartment, coming from the resorbed informal housing settlements. [29] In the cite de transit, the administration believed strict discipline was necessary to "initiate families to modern life"[30] and allow the transit to HLM housing. Inadaptation was not the only reason to house Algerian families in *cités de transit*: families from the informal housing settlement were frequently put in the centres because there was simply not enough HLM housing available (Cohen, 2013). Most of the *cités* were managed by the Sonacotra or the Cetrafa, an organization that stemmed from the colonial period. Because of the lack of other housing options and HLM availability, North African families were housed in these centres for on average eight years, even though they were supposed to be transitory and *educatif* [educational] (Zehraoui, 1976). By putting families in *cités de transit*, sometimes for long periods of time, the French administration marked their racialized difference, precariousness, and their inassimilability while at the same time arguing that these practices were necessary for assimilation. The presence of French wives thus upset this racialized difference that allowed the French administration to regulate North Africans through colonial modalities of governance.

The social workers and HLM bureaucrats determined who was considered adapted to live in HLM housing, whereby they measured "adaptability" by looking at the behaviour of the wife.[31] Similar to the colonial context, the social action interventions in both the *cités de transit* as well as the social action services in the HLM targeted almost exclusively women. There were no actual clear criteria on what basis these measurements were made: rather, the measurement of *adaptation* was considered to be self-explanatory, based upon colonial knowledge on the Muslim population. However, the administration did not administratively or politically acknowledge the existence of households in which the wife was white French or European. In the policy discussions on housing and social action policies, the existence of French wives is invisibilized.

In the process of measuring so-called adaptability, authorities argued that interracialized households were more "adaptable" to the French lifestyle compared to mono-racialized North African families. For example, the Préfet de la Seine-Maritime wrote in 1960 that the housing authorities regarded European wives more favourably to manage a household than Muslim women.[32] Accordingly, of all 205 families categorised as "North African", 30.7% of "mixed families" in Le Havre live in HLM, and only 18% of 'Muslim' families live in HLM. Moreover, the ESNA for example shows that in their research from 1963, 17 families North African live in HLM housing, of which seven are "mixed".[33] The sample of HLM housing provided in David's research on the communist working-class city of Saint Denis that four out of nine families categorized as "Algerian" were actually interracialized (David, 2016). This shows consistently over sources an overrepresentation of "mixed" couples.

The authorities did not consider the presence of white women as an indication that the strict hierarchization of assimilability was not tenable. This difference was motivated by the idea that European wives were better at keeping a proper household, as the Préfet de le Seine Maritime articulated in a letter to the *Service des Affaires Musulmanes*.

> If mixed households seem to have an advantage in this distribution, it is because, most of the time, these European spouses benefit from a favourable prejudice in the keeping of the household, in relation to Muslim women, who need in most cases, without being inferior, an adaptation period to the western life more or less prolonged.[34]

The state officials do not recognize the irony that they were measuring European women's (frequently French) assimilation into "western life", because these women were

subsumed within the North African population. By measuring "adaptability" the authorities could include interracialized couples in the racialized hierarchy of assimilability without complicating the separation between white and non-white families.

The categories on which the housing policies for families were based classified the household as a pre-existing uniform patriarchal racialised category, consisting of a male head of household, and a dependent wife and children. By subsuming interracialised families within the racialised migrant population and not acknowledging the presence of white women or the possibility for interracialised household, the "family" remained a racialised monolithic category.In doing so, the authorities thus did not acknowledge interracialized families as salient to their understanding of assimilation, of the tolerance thresholds, of cohabitation policies, or of segregationist measures. The negation of interracialized households allowed the administration to either segregate or enforce cohabitation policies for these racialized households categories that ostensibly promoted assimilation.

## 7.    Preventing interracialized intimacies

Where housing for families perpetuated racialized boundaries in urban and domestic space by negating interracialized households, the housing for single men perpetuated racialized boundaries by isolating North African men. Hierarchies of gendered assimilability tjustified exclusion of North African migrants from the French community. These continuously necessitated the reinforcement of racialized boundaries within the domestic space. To do so, administration also prevented interracialized intimacies to form in the first place.

The administration mandated the Sonacotral (renamed SONACOTRA after Algerian independence) to build and manage hotel-type of housing known as *foyers* for single North African men. These had an unspecified legal category and were meant to offer sanitary housing in a cost-efficient and regulated manner, that would break down the "tribalism" that reigned, according to officials, in informal housing. The foyer was a collective housing structure, but at the same time it highly individualized and isolated its

residents, because community ties between the residents was made difficult by surveillance and the lack of feeling of ownership of the spaces (Sayad 1980). The resident had no renters' rights and hence was stuck in an in-between space: he has a bed, not a home, thereby producing temporariness and precarity (Hmed, 2006). By making the foyer the only type of housing available (besides informal dilapidated housing), the French administration could keep single men outside of the French community, both spatially and intimately.

The administration motivated the necessity to house migrant men separately out of concerns for their inassimilability. Mixing housing for single Algerian men with housing for Algerian families was considered in the 1960s, so before Algerian independence, but never actually realised. When in 1960 the Prefect of the department in the Alps wanted to mix Algerian families and single men in one housing facility, the Head of Social Affairs of the Algerian Affairs department answered resolutely: "I am very opposed to this project", without further motivating it. In 1961 the social action fund denied a project of the Sonacotral to build a residence with both single men and families.

> After deliberation, we have not been able to approve this project, because we do not consider it desirable, from a social and familial point of view, to group families and single men together in one building. Such a formula does not seem to benefit the evolution of the Muslim family.[35]

The offcicials did not further explain this undesirability. Rather, it was considered evident that single men are corruptible factors in the encouragement of assimilation of families. Single (North) African men could never attain assimilation because they were deemed incapable of being integrated into French domestic life. However, the separation of North African men from families enabled the French authorities to regulate interracialisation.

The administration did not want to build other types of housing for African male migrants. Foyer housing was not exclusive for North African migrants. Still, other groups that were housed in foyers, such as students and young workers, did have other housing options, and only lived in a foyer for a specific period of their life (e.g. during their studies). African migrants only had the private housing market as an alternative option - often either inaccessible to migrant workers or of poor quality.[36] In contrast, the administration did not make an absolute distinction in housing between families and single men for European migrants. A letter from the Préfet de la Seine to the Minister of Interior from

1963 sets out the "problem" of housing for workers from "African states".[37] The Préfet argued that both Sub Saharan and North African workers need to be housed separately from families. He acknowledged that Spanish and Portuguese migrants also have housing problems. Yet, he argued that rather than foyers, regular housing had to be constructed for these groups without explaining why. This unexplained self-evidence illustrates how the hierarchies of assimilability motivated inclusion into the national home of European men and exclusion of African men.

The housing policies for North African men were based on the construction of threat posed by North African men. Heyman argues that collective distrust of categeories of people focuses the attention of the state: they become hypervisible as "risks" (Heyman, 2009). Racialisation plays a role in collective distrust by state institutions and officials, which in its turn leads to further marignalisation and exclusion (Doyle, 2007; Heyman, 2009). For North African men, institutional distrust was structured by sexualized fears for interracialized encounters. Looking at spatial regulation of the foyers and the internal regulations together shows that housing for single men discouraged any mixture with the white French population, specifically with white French women.

The authorities used segregationist policies to keep North African single men spatially distant from the white French population. Mayors refused to build foyers in their municipality, invoking arguments on the danger that single North African men (supposedly) pose to (white) women and young girls. Massenet, the head of the Social Action Fund and on the board of the SONACOTRA, proposed in a speech that spatial dispersion of North African men was necessary because "women and young girls dare not leave the house because they fear they will be attacked and raped" (Shepard, 2018, p. 233). Moreover, residents of neighborhoods protested the building of foyers because they feared it would pose threats to public order.[38] However, the social action service for migrants reported in 1971 that no instances of threats against women have occurred.[39] This indicates that sexual anxieties were based on collective distrust rather than instances of sexual violence.

Research has argued that the logic of surveillance explicitly underlined the building and working of foyers (Bernardot, 2008; Hmed, 2006). As these men were seen as uprooted and potentially dangerous, strict internal rules applied in the foyers. The SONACOTRA

recruited the *gérants* [managers/concierges], who were responsible for the inner workings of the foyer, mainly amongst the colonial officers previously in Algeria, for they were thought to "know the people". And so, the officers who had been enforcing colonial rule now enforced the rules in the foyers. The Sonacotral was directed by Jean Vaujour, who had been the architect behind the forced displacement of rural communities in Algeria. In describing Sonacotra's mandate, Vaujour explicitly referred to the housing projects in colonial Algeria and vowed to make the Sonacotra foyers places for "moral and sanitary progress" (Bernardot, 2008, p. 48). In 1965, the Préfet of the Ain region motivated building a new foyer by arguing that "it would also be suitable to achieve better police surveillance of the hostel itself and its surroundings, surveillance for which its current location does not lend itself well".[40] Sources that give the perspective of migrant men show that they were aware and attempted to defy such surveillance: for example, single men who were expulsed from the informal shantytowns did not want to be rehoused in foyers because of the widespread police surveillance.[41] The housing structures enabled police surveillance that was integral to the control of migrant workers, which still today enables policing in the *banlieues* (Jobard, 2020; Rigouste, 2014) .

Through this logic of surveillance, the SONACOTRA strictly controlled the possibilities for intimacy through communal living and internal rules that prohibited outside visitors. As illustrated in the right-leaning newspaper *Le Figaro,* the foyers ensured the residents were controlled like children in a boarding school: "no visits in the foyer. Those are the rules! Even though they are past the age of boarding school."[42] This affected the possibilities for intimacy for the residents. Moreover, a research carried out in 1960 amongst Algerian workers shows that half of the respondents wanted a single room. One of the reasons given is the desire to have (a) romantic and/or sexual partner(s).[43] This illustrates that through the structure and the internal regulations, the administration could regulate the possibilities for a private life, and ultimately, the domestic life of the foyer residents. Analysing the prohibition of visits together with the sexual anxieties that motivated segregation shows that this prohibition was partly aimed at the prevention of interracialized intimacies.

The right to visitation was an essential question in the internal management of the foyers. The former "director of research and programming" of the SONACOTRA stated in an interview that "this was the big issue at the time: visits, especially female visits"

(Bernardot, 2008, p. 126). Female visits were considered a self-evident problem, as illustrated in the research on migrant housing carried out by the SLPM from 1971.

> There is another problem that is continuously relevant. It is demanded by the residents, and even more by those who want to be their spokespeople, or even their defenders, towards the exterior, is the problem of the right to visitations and its limitations. […] We comprehend that some, or even many, want the right to visitors to be extended to women (lit. female sex) and exercised within the rooms. […] But *evidently* the problem of meetings with women remains [emphasis by author].[44]

The problematization of female visits shows that the authorities worried about intimate heterosexual relations. However, they did not explain why it would be a problem to have "meetings with women", considering the desire to prevent interracialized intimacies in the foyers self-evident.

The desire of the authorities to ensure that single men would not have intimate relations with white women revolved amongst others around anxieties about sexual relations between men and (white) French women, based on the reactivation of old colonial stereotypes of Muslim men as non-sociable sexually violent men (André, 2016; Brun & Shepard, 2016; Ruscio, 2016). Stereotypes on sexual danger and deviant sexual and gender norms still today structure anti-Muslim stereotypes, and have only slightly decreased in the last thirty years (Yuma, Mayer, Michelat, Tiberj, & Vitale, 2020; Ticktin 2008). In the colonies, interracialized intimacies had the potential to upset the colonial hierarchies (Stoler, 1989). Interracialized intimacies can upset racial categories, as categorisations are threatened, but thereby also solidified, on their boundaries. In the 1960s and 1970s in France, such intimacies did not fit the paradigm of assimilation that excluded North African men and problematized North African families. And thus, the foyer-housing enabled the authorities to prevent interracialized intimacies to reinforce and justify racial boundaries. Collective distrust justified surveillance and segregation, which reinforced marginalization of North African families.

## 8.    Conclusion

I have argued the French administration employed colonial practices to regulate the presence of migrant from North Africa in the hexagon through housing policies, that brought together the regulation of urban space with the domestic space. By tracing

fragmented information in the government archive, and informed by feminist research on the regulation of intimacy and domesticity, I was able to retrieve invisibilized presence of interracialized households. Focusing on the presence of interracialized households and the regulation of interracialized intimacies in the analysis of housing policies, I have shown how the housing policies were justified by hierarchies of assimilability that were dependent on the negation and prevention of interracialized intimacies and households.

This paper has argued that housing policies for families encouraged spatial distribution based on an understanding of the racialized difference between North African and white families. The administration did not acknowledge interracialized households as transgressing the categories on which cohabitation and segregation policies in the urban space were based. Moreover, I have argued that housing policies enabled the authorities to segregate single migrants and intervened in their intimate lives in a way that it did not allow for the construction of domesticity, which worked to prevent interracialized intimacies. Looking at the co-production of domestic space and urban space thus shows how gendered assimilation policies for migrant families and gendered exclusion of single men are interconnected through the regulation of interracialization.

Official discussions about the housing policies revolved around assimilation, "evolution", "adaptation", and tolerance. The policies did not have the effect, nor the intended effect, of including the North African population into the French community. Rather, the regulation of domestic and urban space through housing policies functioned as a tool of governance that reinforces the racialized difference between the white French population and the North African population: housing was racial project that consolidated racialized difference. This justified both the separation between North African families and French/European families, and justified the separation between single men and families. Separation of North African migrants enabled interventions in the domestic space to discipline and control North African migrants and to surveil and exclude North African men in the name of gendered assimilation. In doing so, the policies not only perpetuated racial boundaries in urban space, but also racial boundaries in domestic space.

The insights from this paper point to the necessity to look at the cross-analysis of spatial regulation and the regulation of domestic space to more fully understand the production and perpetuation of racial boundaries in migrant housing policies. It shows that housing

policies are racial projects on two levels: on the domestic level and the urban level. This brings a feminist insight to the regulation of housing: racial boundaries are dependent on racialization in the domestic space. This shows that the contemporary discussions on housing and spatial integration of migrants should take the regulation of intimacy and domesticity seriously.

## 9.    Endnotes

[1] Whereas the archives I analyze use "mixed households/families", I use interracialized households to underline the action required to make racialized identity salient, rather than it being a pre-existing reality that 'mixes'.

[2] For discussion on French colonial humanism and the ways in which cultural racism played an integral role in French colonialism, see Wilder, G. (2005). *The French imperial nation-state: Negritude and colonial humanism between the two world wars*. University of Chicago Press.

[3] «Ministre de l'Interieur a Monsieur le Premier Ministre, action sociale en faveurs des travailleurs africains établis en France, 18 June 1963, in Archives Nationales 19770346/10.

[4] Synthese de rapports des charges de mission du service de liaison et de promotion des migrants,  Ministre de l'Interieur, 1966, in Archives Nationales 19770346/10. All from French translations are made by the author.

[5]  Synthèse des rapports trimestriels établis par les conseillers techniques pour les affaires musulmanes, Ministère de l'Interieur. Service des affaires musulmanes et de l'action sociale, confidentiel. 1959-1965, in Archives Nationales 19760133/14.

[6] Etudes sociales nord africaines, compte rendu d'activité association d'entraide nord-africaine d'Indre et Loire année 1961, 30 June 1962, in the private archives of Monique Hervo ARC-3019-11.

[7] Rapport sur l'immigration familiale, direction de population et de migration, Ministere de l'Interieur, 1968, in Archives Nationales 19950493/5.

[8] Rapports Trimestriels du Conseillers techniques pour les affaires musulmane,1958-1963, in Archives Nationales 19760133/14.

[9] Rapport Correart de la CETRAFA SLPM, octobre 1965, Les cités de transit pour famille. Sent to the F.A.S. on 5 October 1965, in Archives Nationales 19770391/6.

[10] For example, one of the reasons/examples given of the assimilability of European migrants is the high prevalence of mixed marriages between Italians or Polish and French people. Synthèse des rapports trimestriels établis par les conseillers techniques pour les affaires musulmanes, ministère de l'interieur. Service des affaires musulmanes et de l'action sociale, July 1967, confidentiel, in Archives Nationales 19770346/10.

[11] For example, the circular of 5 October 1972 (on the "preparation for the resorption of unsanitary housing and for the housing of immigrants) differentiates between "the foyers for isolate workers" and "family housing".

[12] Referred to in French as the "resorption des bidonvilles".

[13] Circular of 19 April 1972 "relative aux cités de transit" stipulates *"obviously the cités de transit are likely to receive foreign families as well as families of French origin"*.

[14] Rapport a Monsieur le Préfet de la Region Parisienne sur la Résorption des bidonvilles et les problèmes des migrants, 1971, in Archives Nationales 19770317/1.

[15] ESNA recherches les africains du nord a Grenevilliers 1963, In Archives Municipale Seine-Saint-Denis ASD-37AC17.

[16].Liberation 11 June 1974, Claude Liscia et Gérard Melchior. In Blanc-Chaléard 2016, p. 357.

[17] Asti, info sur les cités de transit, 1974, , Info sur les cités de transit, personal documents of Monique Hervo (undated), in private archives of Monique Hervo ARC-3019-11.

[18] Personal documents « la cité de transit solution pratique pour le relogement des travailleurs immigrés » by Monique Hervo, (undated) likely 1974, in private archives of Monique Hervo ARC-3019-11.

[19] In Liscia, C., l'enferment des cités de transit, edited by the migrant organization La Cimade, 1977,  in the private archives of Monique Hervo, ARC3019/11. This booklet explains how the gérants of the

centers kept a close eye on the residents and in cooperation with the police, expulsed undesired migrants to Algeria, including children who had never been to Algeria.

[20] Lettre du Prefet de la region Rhone-Alpes, Préfet du Rhone a Monsieur le Premier Ministre, Le Ministre de l'Interieur, le Ministre du Travail, de l'Emploi et de la Population, 5 January 1972, in Archives Nationales 19860269/11.

[21] Lettre du Prefet de la region Rhone-Alpes, Préfet du Rhone a Monsieur le Premier Ministre, Le Ministre de l'Interieur, le Ministre du Travail, de l'Emploi et de la Population, 5 January 1972, in Archives Nationales 19860269/11.

[22] Ibid.

[23] Synthese de rapport des charges de mission du service de liaison et de promotion des migrants, SPLM, July 1967, in Archives Nationales 19770346/10.

[24] Circular 15 June 1970, « limitation de l'admission des familles étrangeres ».

[25] Monique Hervo kept notes on all the families that passed through the bidonville. She has recorded 10 'mixed couples' in the bidonville of Nanterre, out of (about) 210 couples/families. In private archives of Monique Hervo, 1968, ARC/3019/4. I have changed the names to comply with privacy requirements of the archives.

[26] Ibid.

[27] Ibid.

[28] Etudes Sociales Nord Africaines, recherches des africains du nord a Grenevilliers 1963, in Archives Municipale Seine-Saint-Denis 37AC17.

[29] Rapport Correard de la CETRAFA SLPM, octobre 1965, Les cites de transit pour famille. Sent to the F.A.S. on 5 October 1965, in Archives Nationales 19770391/6. This report was compiled by the former CTAM officer Jean Correard.

[30] E.g.  in Etudes « La cohabitation des familles françaises et étrangeres », Ministere de l'Equipement et du Logement, August 1970, in Archives Nationales 19771141/17.

[31]  Rapport Correard de la CETRAFA SLPM, octobre 1965, Les cites de transit pour famille. Sent to the F.A.S. on 5 October 1965, in Archives Nationales 19770391/6.

[32] Préfet de la Seine-Maritime a le Ministre de l'Intérieur, services des affaires musulmanes. Objet : Logement des travailleurs musulmans et leurs familles, 1960, in Archives Nationales 19770391/6 .

[33] Etudes Sociales Nord Africaines, recherches des africains du nord a Grenevilliers 1963, in Archives Municipale Seine-Saint-Denis 37AC17.

[34] Préfet de la Seine-Maritime a le Ministre de l'Intérieur, services des affaires musulmanes. Objet : Logement des travailleurs musulmans et leurs familles, 1960, in Archives Nationales 19770391/6.

[35] Lettre du fonds d'action social au monsieur le directeur de la Sonacotral, 21 July 1961, in Archives Nationales 19770391/6.

[36] In 2021, retired African men still live in foyers, unable to find affordable housing alternatives and without the possibility to go to their countries of origin without losing their pensions.

[37] Prefet de la Seine au Ministre de l'Interieur, conditions d'habitats de ressortissants d'etats africains, 20 March 1963, in Archives nationales 19770346/10.

[38] E.g. Synthèse des rapports trimestriels établis par les conseillers techniques pour les affaires musulmanes, 1e trimestre 1959, ministère de l'interieur. Service des affaires musulmanes et de l'action sociale, confidentiel, in Archives Nationales 19760133/14.

[39] SLPM, Note sur l'immigration etrangere dans le Rhone, 1971, in Archives Nationales 19860269/11.

[40] Préfet de l'Ain, a monsieur le premier ministre delegation de l'action sociale pour les travailleurs etrangers et pour information a ministre de l'interieur, 7 May 1961, objet : hebergements des travailleurs d'outre-mer et notamment des travailleurs algériens., in Archives Nationales 19770391/6.

[41] Etudes de l'habitat, GEANAAP, 1957, in private archives of Monique Hervo, ARC/3019/2.

[42] Le Figaro, 19 janvier 1973, la fin d'un foyer-taudis, retrieved from the archives of the Ministry of Interior that include newspaper clippings on the rent strikes, collected by the director of public liberty and juridical affairs, in Archives Nationales 19960134/3.

[43] Sondage d'opinion publique sur les aspirations des travailleurs en matière de logement, service des affaires musulmanes to fond d'action sociale, June 1960, in Archives Nationales 19770391/6.

[44] Report by the 'service de liaison et de promotion des migrants' by the prefecture de la region Parisienne, for the department of Paris on the 'resorption of shantytowns and the problem of migrants', 1 March 1971, in Archives Nationales 19770317/1.

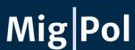

## 10. Archives

AN = Archives Nationales
ASD = Archives Seine-Saint Denis
MH = Archives Monique Hervo

| AN | 19760133 | 14 | Secrétariat du directeur (direction de la population et des migrations) (1966-1980) |
|---|---|---|---|
| | 19770317 | 1 | Chargé de mission (direction des personnels, des affaires politiques et de l'administration territoriale, ministère de l'Intérieur) |
| | 19770346 | 10 | Secrétariat (sous-direction des affaires politiques et de la vie associative, ministère de l'Intérieur) |
| | 19770391 | 6 | Direction de la population et des migrations (1966-1980) |
| | 19771141 | 17 | France. Direction de la Construction (1944-1998) |
| | 19860269 | 11 | Direction de la population et des migrations (1966-1980) |
| | 19950493 | 5 | Direction de la population et des migrations (1966-1980) |
| | 19960134 | 3 | Bureau des étrangers relevant des régimes spéciaux (direction des libertés publiques et des affaires juridiques, ministère de l'Intérieur) |
| ASD | 37AC | 17 | Municipalité Seine-Saint Denis |
| MH | ARC 3019 | 2 | Monique Hervo, Demandes de logements, Attributions, Enquetes, Notes de Monique Hervo |
| | | 4 | Monique Hervo, « Carnet de bord », 1959-1971 |
| | | 5 | Monique Hervo, Nanterre. Dossiers des familles |
| | | 6 | Monique Hervo, Nanterre. Dossiers des familles |
| | | 7 | Monique Hervo, Nanterre. Dossiers des familles |
| | | 11 | Monique Hervo, Cités de transit en France |

## 11. Acknowledgements

This research was funded by the European Research Council, as part of the project 'Regulating Mixed Intimacies in Europe (EUROMIX)' [grant number 725238]. I am grateful to the anonymous reviewers and editor, whose valuable and constructive feedback have been of great help. I would like to thank the Senior Editorial Fellows: Dr. Saskia Bonjour, Dr. Evelyn Ersanilli, and in particular, Dr. Darshan Vigneswaran for their feedback and for organizing the migration politics writing residency. I would also like to thank the migration politics fellows and all commentators who provided such valuable feedback, in particular Dr. Stephan Scheel, Dr. Anja Karlsson Franck, Prof. Loren Landau, and Dr. Wouter van Gent.

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
