# Peer review of "(Inter)racialization: the regulation of domestic and urban space in housing North African migrants in post-colonial France"

_Migration Politics_

## Round 1 · Referee Report · Anonymous · 2021-7-16

Report

The article continues the exploration of a central ambivalence in French housing policies: while claiming to promote integration and social mix, they actually reinforce segregation. Even more importantly, far from being color-blind, they rest on and create racial boundaries. The goal of the author is to add an element to the significant set of works that have brought into light this phenomenon and its evolution from colonial rule in Algeria to contemporary urban policies promoting “mixité sociale”. Focusing on housing policies in the 1960s and 1970s, the author argues that not only do they shape urban space, but they have an impact on the domestic sphere. More generally, referring to the colonial project which, at the same time, built housing and crafted domesticity, s/he calls for a perspective that take into account the private and the public to fully understand the issues and outcomes of these policies.

This perspective is undoubtedly valuable and extremely stimulating.
The author develops an argument based on a solid bibliography, studies on housing and urban policies as well as on colonial history.

The article is very well written, clear, fluid, rigorous.

My reservation comes from the development of the argumentation that, in my view, brings together heterogenous points: the representation of migrants as single men, their exclusion from the French community and their relegation in “foyers de travailleurs migrants” form one and compelling argument; then the author addresses the issue of gender and explains how women have been the main targets of assimilation policies; the most novel point is the administration of interracialized couples and how it prevents “interracialized intimacies”; section 5 about cités de transit and the combination of segregation and distribution (through “seuil de tolerance”) provides less original results.
In sum, although the author claims to make a significant contribution, and even coins a new notion (“sterilized assimilation”), s/he could be clearer about what is really original and what is built on existing research.

Requested changes

See Report

---

## Round 1 · Referee Report · Anonymous · 2021-7-23

Report

I was generally impressed with this manuscript. The author discusses the colonial continuum as represented in housing policies targeting North Africa migrants after decolonization, thereby focusing on the domestic sphere as a site for assimilation. Through archival research, the author tracks how migrant families were designated assimilable or not. I appreciate the incorporation of “interracialization” as a way to make sense of the racialization in the domestic sphere. I would suggest augmenting this intervention by discussing exactly how it contributes to extant scholarship, particularly re racialization in France. The author also details how the state manage and controls race and gender boundaries, and how assimilation is constructed along racial and gendered lines. Regarding the role of gender in discourses on assimilation, I would suggest consulting Nimisha Barton’s book, Reproductive Citizens. I would also suggest making the arguments in this paper more strongly, for example by stating how housing policies are an example of racial project (Omi and Winant 1994) in a seemingly non-racial society. I also wanted more specificity regarding the archival analysis procedure. I like that the author emphasizes the archives both as objects of knowledge and the importance of reading along the archival grain, but that also made me curious about exactly the analytical procedures the author used.

In terms of writing style, etc., I would suggest structuring the article around a particular argument and making clearer the intervention into extant literature. To this end, I think the least convincing element of this article is the term “sterilized assimilation.” It actually does not feature in the manuscript as much as the title and abstract might suggest, and as written, it does not add much substantively to the manuscript. As a concept, “sterilized assimilation” is not really explained or developed (and I would more generally argue that assimilation is in general more about exclusion rather than inclusion, so as written, it’s unclear what how this understanding of assimilation is different). Towards the end of the article, the author references the contradiction of social mixite with marginalization and segregation, but really this is not a contradiction, especially in the context of France. I would suggest making the point a bit more strongly – that race becomes activated or salient in the same between the two.

---

## Round 1 · Referee Report · Anonymous · 2021-8-31

Report

Overall, this is a conceptually and empirically rich paper and I really enjoyed reading it. I was especially excited by the depth of the analysis and the way the authors frame how authorities have reinforced racial boundaries rather than encouraging inclusion.

Historical sources are very rich and the authors are very effective in coping with both description and explanation.

And the paper is very well written, even if English style may be improved to avoid a French structure to sentences. 

That said, there is some work still to be done to bring so much material together in a cohesive manner. 

My comments and suggestions are just to help the authors to improve their argument and better prove their main results.

Concerning the abstract: your beautiful paper deserves a better abstract. Your current abstract is not attractive. I suggest rewriting it and clearly state your theoretical framework, your sources, archive, main results, the gap in the literature you are able to fill. At the moment, is too generic, just pointing the topic, and not your main discoveries and results, and why it is new for the literature on housing and assimilation.

Same concerning the main argument, related to the title of your paper: you point to the relation between domestic and urban space. It is a pity that you develop much more the production of domestic space in housing African migrants, while concerning the production of urban space you just relate to segregation. Maybe you can develop in a deeper way using your archive sources.

Concerning sterilized assimilation: you miss a precise definition. Due to the fact that you too frequently relate to the current debate, you tend to avoid defining what was “assimilation” like in the Sixties. Also you may better relate the adjective “sterilized” to its intellectual frame.

There is probably a point related to trust and distrust of French authorities that could be better developed. Probably you can mobilise classic work by Weber and Schluchter. Trust conceptualized in a Weberian manner, is strictly related to conventional forms of link between individuals, procedures, and institutions. Trust in migrants and their family-making practices is not structured by impersonality or regulation, but by the lacking of proof of recognition.
I am convinced of the relationship between housing authorities and the regulation of inter-racialized intimacies. It is certainly an important analytical point and one that leads to results. Would not be the case for you also to insist on this point in the introduction, instead of introducing current debates in French political contention?
Please add a table with sources and archive information.

Page 12 line, you may better develop why temporary housing and cite de transit had a repressive character.

Page 14: seems here you do not refer and quote the important results from Nonna Mayer on Islamophobia and anti-Muslims prejudice in France in the long run, concerning the enduring dynamics of strong prejudice against Muslims in French society.

Page 15, are you sure of using “type of inclusion” to talk about mechanisms reinforcing racial boundaries?

Page 10 and following. Could be useful to remember that in the sociology of institution and state agencies there is a principle of symmetry so that we study both what is done and what is not done (or avoided) and not only what is done. See King & Le Galès for Oxford University Press 2017, and in particular the chapters 18 and ch. 22 on assimilation and security, related to your argument).

Page 13: why do you talk about “crafting racial balance”? If you say that racial boundaries are based on segregation and concentration, could you better explain what is the role of balancing in the producing of sterilized assimilation? Maybe you can be more precise on the effect of this strategy?

With regard to the historical relation between police and migrant working class, it seems to me that it is important for your work to examine Policing the Banlieues, by Fabien Jobard, in Policing in France ISBN9780429026928

Overall, this is an important subject and could become an excellent article, but the current version would benefit some more clarity and much elaboration in dealing with the powerful linkage between the theoretical and the empirical parts of the research. 

Overall, I thought this is an important, deep, and generative article and I hope the above comments are helpful to the authors.

Requested changes

See Report

---

## Round 2 · Referee Report · Anonymous (Referee 3) · 2022-1-1

Report

This is an important article, and I am very honoured to evaluate the second version of the paper. Without a doubt, the article should be published as soon as possible.

Still, a little bit of work is needed, but for sure, it does not require any substantial revision work in its structure.

I’ve appreciated a lot the work done on the paper!
The abstract is much clearer, even if there is a typo in its last line.

The new introduction is better than the previous one and more focused. It helps the reader and avoids redundant conceptualisations. I consider it essential, sorry to insist, to include the trust dimension. I'm afraid I disagree that including a precise language on trust will require archive work, but just more theoretical precision. Your paper deals with trust dimensions, and it is a pity not to treat them with a precise conceptual language. Especially in a feminist framework, how the state trusts or mistrust categories of individuals is a foundational analytical dimension. It could be quick to do, looking at the Weberian legacy to cope with the trust/mistrust issue in the section on the regulation of interracialised intimacies through the housing policies targeting single men. It seems that you can better specify the basis of interracialised intimacies regulation: aren’t they based on mistrust and a sense of threat?
It also seems that your main argument related to the fact that in France, race becomes salient in the context of social mix & marginalisation requires a bit of reference to the rich literature in urban planning and, broadly speaking in French urban studies: could be relevant to reinforce your argument to relate to Clément Boisseuil important paper Boisseuil, C. Governing ambiguity and implementing cross-sectoral programmes: urban regeneration for social mix in Paris. J Hous and the Built Environ 34, 425–440 (2019). https://doi.org/10.1007/s10901-019-09644-4. Equally, you can relate to Christian Lefèvre et al. contribution on how the same way of governing housing policy in the suburbs remains related to the kind of regulatory criteria you analysed for the ’50s and ’60s (see "Les défis de la gouvernance métropolitaine." (2013): pp. 21-34).
I appreciate your reference to the work of Fabien Jobard, which was missing in your paper. Still, your choice of looking at the same time at what the government has done and what has omitted to do is essential in studying local regulation of immigration. I suggest making your choice more explicit and referring to chapter 22 on immigration and assimilation in the most recent book of historical-comparative sociology of the state in Europe (Reconfiguring European States in Crisis, edited by Desmond King and Patrick Le Galès).
Last but not least: concerning the reactivation of old colonial stereotypes of Muslim men as non-sociable sexually violent men, it would be interesting for you to reinforce your argument relating to the work of Nonna Mayer & al. showing how, in the last 30 years, the stereotypes about Muslim men have only slightly decreased in France: you can easily refer to the open edition of Mayer et al. 2021 long contribution to the cncdh report: "Mise en perspective de trente ans d’évolution par les chercheurs.", pp 33-111.

Overall, this is an excellent article. It is an important, profound, and generative article, and I hope the above comments are helpful to the author.

---

## Round 2 · Referee Report · Anonymous (Referee 1) · 2022-1-20

Report

Yes.

---

## Round 2 · Referee Report · Jean Beaman (Referee 4) · 2022-2-9

Report

This is an excellent revision of this manuscript. I really appreciate how the author responded to the reviewers' comments and suggestions. The manuscript is much stronger being explicitly framed around interracialization, and the explicit bringing together of regulating domestic and urban space. By doing so, the author provides more of an intervention in extant literature which will be useful to other scholars. The topic and framing of the manuscript has much more "portability." Moreover, the thesis and overall argument are much clearer – “I look at how regulation disposes a racialized order as the natural order of things” (p. 4). I also appreciate the attention to how the author thought through their methodological choices, and how they use the archive. I look forward to citing this manuscript.

---

## Round 2 · Author Response

Dear esteemed colleagues,

I would like to thank the reviewers and editor for their time and highly constructive and valuable criticisms, and the opportunity to send in a revised manuscript.

As the reviewers and editor will find, I have made substantial changes in the structure of the article to incorporate the feedback. In general, I have made the argument on the regulation of interracialized households and intimacies more central to the analysis. I have also moved away from the discussion on contemporary politics of social mixture, and instead, focused on a contextualized discussion of assimilation to explain the regulation of interracialisation. Please see the list of changes for a point-by-point list of all revisions I have made to the manuscript.

I hope that you will find the revisions and the point-by-point changes satisfactory, and I look forward to your response,

Kind regards,

The author

---

## Round 2 · List of Changes

Dear colleagues,

Below, I outline how my revised article incorporates the responses to the issues listed in the reviewers’ reports and editor’s report asking for revisions of the text. I will list below point-by-point changes I have made to the manuscript, based on the editor’s reports and the reviewers’ reports.

1) All reviewers point to a need to make the main argument more visible. They suggest making for example the incorporation of “interracialization” into the analysis more central to the presentation of the argument, especially by discussing how it contributes to existing research (see review 1). This could be the point of entry for the article, and would allow you to address issues of gender and spatial and racial segregation as well as the issue of French wives from this starting point (see review 2).

In response to this valuable suggestion, I rewrote the introduction to centre interracialization and interracialized intimacies instead of focussing on contemporary debates on “mixite sociale” in France (see page 1-2). I moreover restructured the paper to bring the issue of interracialisation more central to the analysis. To do so, I have introduced the topic of interracialized households earlier in the paper (page 8). Moreover, I have brought together under one section the regulation of interracialized intimacies through the housing policies targeting single men (page 14-16). This also attests to reviewer 3’s helpful suggestion to bring the regulation of interracialized intimacies more into focus.

2) All reviewers also point that, while they were curious as to the notion of “sterilized assimilation”, the notion a) lacks a definition; b) does not feature so prominently in the paper. My suggestion would be to question whether you truly need this expression to make your point in the article. The notion of interracialization, or the cross-analysis of domestic and urban spaces both seem to bring enough material to your argument.

Following the helpful comments on the insufficient development of the concept of “sterilized assimilation”, I have taken out this concept and instead focused on the developed of the cross-analysis of domestic and urban space to look at the regulation of interracialized intimacies and households.

3) This also relates to the need for a contextualized definition of assimilation. As such, the paper is sometimes characterized by anachronistic analyses (see review 3), mixing contemporary debates or definitions with the context of the 1960s and 1970s.

Following this important feedback, I have re-shifted the focus from contemporary debates on mixing to interracialized intimacies and interracialization. To do so, I discuss the meaning of assimilation more elaborately to strengthen the argument on why interracialized households and intimacies are negated and prevented. To this end, I have incorporated a new section on “hierarchies of assimilability” in which I set out a contextualized meaning of assimilation, rooted in colonial logics (page 6-8). Subsequently, I show how the housing policies regulate interracialized intimacies and households to consolidate hierarchies of assimilation. In doing so, I have also tried to respond to reviewer 2’s reservation on the bringing together of heterogeneous points of, on the one hand, the regulation of single men in foyers and, on the other hand, gendered assimilation.

4) Finally, this also connects to a need for more precision in the historical perspective you use. Review 1 underlines the need for more detail on the analytical procedures used in the archive. I would add that it would be very useful to know which period you study and discuss exactly, as “post-colonial” seems a rather large time-span.

Following the important point on the necessity to bring precision to the historical and methodological approach, I have elaborated on the archives and archival theory in the methods section, to better explain how I came to my material and the necessity of critical archival methodology.

5) reviewers suggest references that might be useful.

I would like to thank the many helpful reference suggestions, and have included many of the suggestions.

In addition to the editor’s report, I would like to respond to the addititional and more specific feedback in reviewers’ reports.

  1. I would also suggest making the arguments in this paper more strongly, for example by stating how housing policies are an example of racial project (Omi and Winant 1994) in a seemingly non-racial society.

Following reviewers 1 suggestion to understand the housing policies as “racial projects”, I have included Omi and Winant’s understanding of racial projects on page 1.

  1. Towards the end of the article, the author references the contradiction of social mixite with marginalization and segregation, but really this is not a contradiction, especially in the context of France. I would suggest making the point a bit more strongly – that race becomes activated or salient in the same between the two.

I agree with reviewer’s 1 remark on that mixture and marginalization are not a contradiction, and have taken this analylsis out of the manuscript.

  1. you may better develop why temporary housing and cite de transit had a repressive character.

I have expanded on the repressive character of the foyer (page 15 ) and cite de transit (page 10).

  1. There is probably a point related to trust and distrust of French authorities that could be better developed. Probably you can mobilize classic work by Weber and Schluchter. Trust conceptualized in a Weberian manner, is strictly related to conventional forms of link between individuals, procedures, and institutions. Trust in migrants and their family-making practices is not structured by impersonality or regulation, but by the lacking of proof of recognition

Whereas I value this insight, I believe this would necessitate another reading of the archives. I have included the argument on the lack of recognition (page 14) to underline how this enabled the authorities to segregate and ‘mix’ migrant families under the guise of assimilation.

  1. Concerning the abstract: your beautiful paper deserves a better abstract. Your current abstract is not attractive. I suggest rewriting it and clearly state your theoretical framework, your sources, archive, main results, the gap in the literature you are able to fill. At the moment, is too generic, just pointing the topic, and not your main discoveries and results, and why it is new for the literature on housing and assimilation.

Following this valuable feedback, I have completely rewritten the abstract to better showcase the results and its contribution to the literature.

  1. Please add a table with sources and archive information

I have added a table to provide the necessary information on the archival sources.

Again, I would like to thank the reviewers and the editor for the helpful and encouraging feedback that greatly helped improve my manuscript.

Kind regards,

The author

---

## Round 3 · Author Response

Dear reviewers,

I would like to thank you for engaging with my work and for your generous feedback and comments throughout this process. As for the latest revisions, I have made the following changes:

1. I have corrected the typo in the abstract and have generally corrected some additional typos, formatting and textual issues.
2. I have incorporated the proposed literature in the anonymous reviewer 1, whom I would like to thank for the suggestions.
a. I have incorporated the research by Lefevre and by Boisseuil on page 1 in the first paragraph.
b. I have added the point on the necessity to study what the state does and does not do on page 5, using King et al.’s work.
c. I have added a paragraph on trust on page 20, to further specify the authorities’ mistrust of North African migrant men.
d. I have incorporated Mayer et al.’s research on page 22.

I hope that you will find the additional revisions satisfactory and would again like to thank for your time and effort into helping me improve this paper.

Best,
The author.

---

## Editorial Decision

unknown